# Dynamic Adjustments of Parenteral Support in Early Adult Intestinal Failure—Essential Role of Sodium

**DOI:** 10.3390/nu12113426

**Published:** 2020-11-08

**Authors:** Torid Jacob, Aenne Glass, Maria Witte, Johannes Reiner, Georg Lamprecht

**Affiliations:** 1Division of Gastroenterology and Endocrinology, Department of Internal Medicine, University Medical Center Rostock, Ernst-Heydemann-Str. 6, 18057 Rostock, Germany; torid.jacob@med.uni-rostock.de (T.J.); johannes.reiner@med.uni-rostock.de (J.R.); 2Institute for Biostatistics and Informatics in Medicine and Ageing Research, University Medical Center Rostock, Ernst-Heydemann-Str. 8, 18057 Rostock, Germany; aenne.glass@med.uni-rostock.de; 3Department of General, Visceral, Vascular and Transplantation Surgery, University Medical Center Rostock, Schillingallee 35, 18057 Rostock, Germany; maria.witte@med.uni-rostock.de

**Keywords:** intestinal failure, parenteral support, adaptation, reconstructive surgery, sodium, BMI

## Abstract

Intestinal failure (IF) requires parenteral support (PS) substituting energy, water, and electrolytes to compensate intestinal losses and replenish deficits. Convalescence, adaptation, and reconstructive surgery facilitate PS reduction. We analyzed the effect of changes of PS on body mass index (BMI) in early adult IF. Energy, volume, and sodium content of PS and BMI were collected at the initial contact (FIRST), the time of maximal PS and BMI (MAX) and the last contact (LAST). Patients were categorized based on functional anatomy: small bowel enterostomy—group 1, jejuno-colic anastomosis—group 2. Analysis of variance was used to test the relative impact of changes in energy, volume, or sodium. Total of 50 patients were followed for 596 days. Although energy, volume, and sodium support were already high at FIRST, we increased PS to MAX, which was accompanied by a significant BMI increase. Thereafter PS could be reduced significantly, leading to a small BMI decrease in group 1, but not in group 2. Increased sodium support had a stronger impact on BMI than energy or volume. Total of 13 patients were weaned. Dynamic PS adjustments are required in the early phase of adult IF. Vigorous sodium support acts as an independent factor.

## 1. Introduction

Intestinal failure (IF) is a clinical syndrome in which intestinal absorption of macronutrients and/or water and electrolytes is diminished to a degree, that intravenous supplementation is necessary to maintain health or growth [1]. Most often it is the result of short bowel secondary to intestinal resection(s) [2]. After resection intestinal adaptation slowly ensues as a compensatory process which includes structural, functional, and behavioral mechanisms that result in enhanced net-absorption, hyperphagia, and reduced caloric needs due to lower energy requirements [3,4,5]. Besides from symptomatic treatment and nutritional interventions, reconstructive surgery helps to achieve oral autonomy. The recently approved GLP-2 analogue, teduglutide, boosts adaptation by mucosal hyperplasia and other, less well-understood mechanisms [6,7,8].

With regard to macronutrients the composition (“compounding”) of parenteral support (PS) is guided by the estimated energy and amino acid requirements, the nutritional status and the course of body weight [9]. Regarding water and electrolytes, it is tailored based on an adequate urinary volume as well as normal serum electrolyte values [9]. Nevertheless, a structured algorithm for the optimization of parenteral support to provide enough of each component, but not more than enough, has not yet been developed except for the volume of PS in the setting of clinical trials testing the effect of teduglutide [7,10]. This is of note because the overall goal is to reduce PS, achieve infusion free days, and aim for oral autonomy [9].

Adaptation can be qualitatively categorized as achievement of oral autonomy versus dependence on PS. Based on that, the different prognosis and the different requirements for fluid and electrolytes of type-I functional anatomy, i.e., end-jejunostomy, versus type-II and type-III functional anatomy, i.e., jejuno-colic and jejuno-ileo-colic anastomosis, have been worked out and the different time courses to achieve oral autonomy have been described [2,11]. But quantitative changes in the composition of PS along the time axis have not been studied in the early phase of adult IF. Therefore, we have analyzed the dynamic changes in the composition of PS, in order to characterize the specific effects of some of its principal components, namely energy, volume, and sodium on BMI. BMI was chosen because it clinically reflects the cumulative effect of PS, convalescence after surgery and intestinal adaptation to the short bowel situation.

## 2. Patients and Methods

### 2.1. Patients

All patients managed by the intestinal failure outpatient clinic of the University Medical Center Rostock between 1 July 2012 and 30 June 2018 were retrospectively analyzed from a prospectively maintained database as of 30 June 2018. Patient characteristics were recorded as suggested by ESPEN [1]. Data were collected in a customized database (Microsoft Access, 2010). The study (German Clinical Trials Register: DRKS00021085) was approved by the local ethics committee according to German law (A 2018-0128).

Parenteral support was prescribed according to the guidelines issued by ESPEN in 2015 [9]. Care was taken to provide enough water and electrolytes to balance fluid losses by ostomies or by diarrhea. This was guided by 24-h urine output of 15–20 mL/kg as well as by serum values within the normal range for sodium, potassium, calcium, magnesium, and phosphate. Energy and amino acids were administered in sufficient amounts to counteract malnutrition, that may have been present when a patient was initially seen. Recompensation of body weight was a treatment goal, although we did not specifically aim for the preexisting weight. Patients were considered malnourished if they had a BMI of less than 18.5 kg/m^2^ [12]. Parenteral support was adjusted based on regular urine collections (including urine sodium in some patients) and interpretation of appetite, general wellbeing, course of body weight, serum albumin, serum urea, gGT and ALT. In addition, phase angle was collected from body impedance analysis (BIA) (Nutribox, Data Input, Pöcking, Germany) when available. All patients received proton pump inhibitors and in cases of very high stomal output this was given intravenously [13]. Antidiarrheals (loperamide, tinctura opii and bile acid sequestrants (if colon was in continuity)) were prescribed based on clinical judgement and patient’s tolerance. Dietary counseling was provided with special emphasis for controlled thirst and the avoidance of excessive drinking, and to keep oral osmotic load low.

Patients were usually seen at the outpatient clinic 4 and 8 weeks after initiation of PS and every 12 weeks thereafter.

Duration of intestinal failure was calculated from the initiation of PS even if it had been initiated at another institution before the patient was seen for the first time in our outpatient clinic. Details of the parenteral support were taken from the prescriptions issued by the compounding pharmacy. The reported values summarize the entire PS.

In order to address the individual time courses of BMI and the individual parenteral support we compared three time points: the first contact of the patient with our clinic (“FIRST”), the time points of maximal PS or maximal BMI (“MAX”), and the last contact (“LAST”). When necessary MAX is indexed as MAX_PS_ for the time of maximal PS or MAX_BMI_ for the time of maximal BMI in the text.

We analyzed the entire cohort and three subgroups based on the functional anatomy described by Messing [2]: patients with a small bowel enterostomy—group 1; patients with a jejuno-colic anastomosis—group 2; and we combined patients with jejuno-ileocolic anastomosis and patients with no prior bowel resection in group 3. Total of 11 patients of group 1 underwent reconstructive surgery after MAX and 2 patients of group 1 received teduglutide (0.05 mg/kg) after MAX. They are not included in the analysis at LAST nor in the comparison between MAX and LAST because both interventions effect adaptation by their own mechanisms.

The components of PS were calculated as the total weekly supplementation divided by 7. BMI was calculated by Quetelet formula in kg/m^2^.

### 2.2. Statistical Analysis

For the description of quantitative variables the median with minimum and maximum and the sample size is reported, and boxplots additionally indicating the mean (as square) are provided. For comparisons of time points (FIRST, MAX, LAST) concerning BMI, energy, amino acids, lipids, glucose, volume, and sodium we used the Friedman-test followed by pairwise Wilcoxon-test and for comparisons of independent groups the Kruskal-Wallis test and Mann-Whitney U-test as non-parametric tests with a *p*-value of <0.05 considered as significant.

The impact of PS on BMI increase was analyzed by the Kruskal-Wallis test followed by Mann-Whitney U-test. For this we considered pairwise combinations of energy, volume, and sodium and show the influence on the absolute increase of BMI between FIRST and MAX. Statistical analyses we performed using IBM SPSS Statistics 25^®^. Figures were prepared using Origin 2018b^®^.

## 3. Results

### 3.1. Demographics

We analyzed 50 outpatients as of 30 June 2018 (Table 1). Of these, 27 had a small bowel enterostomy (group 1) and 16 a jejuno-colic anastomosis (group 2). We combined 2 patients with jejuno-ileocolic anastomosis and 5 patients with no prior bowel resection into one group (group 3). The underlying diseases in group 1 were Crohn’s disease (*n* = 11), complications after other surgical interventions (*n* = 6), ischemia and resection after tumor (*n* = 3 each), resection after ileus (*n* = 2), resection after trauma (*n* = 1) and extensive mucosal disease (*n* = 1). The pathophysiological condition of IF was short bowel in 15 and multiple pathophysiological mechanisms in 12 patients. The median length of the small intestine was 100 cm (min–max: 0–240 cm; *n* = 17; 10 unknown). The length of the residual colon with the potential for reconstructive surgery was 72% (min–max: 29–100%) in 17 patients; 3 patients had no remaining colon and in 7 the length of the remaining colon was unknown. Three patients also had a modified proximal gastrointestinal tract. During the observation period 11 patients had reconstructive surgery. As a result, 4 patients changed to type-II anatomy and 7 changed to type-III anatomy (Table 1).

The underlying diseases of the 16 group 2 patients were Crohn’s disease (*n* = 5), ischemia (*n* = 4), complications after other surgical interventions (*n* = 3), resection after tumor (*n* = 3), and resection after ileus (*n* = 1). The pathophysiological condition of IF was short bowel in 10 patients, mechanical obstruction in 1 patient and multiple pathophysiological mechanisms in 5 patients. The median length of the small intestine was 55 cm (min–max: 10–170 cm; *n* = 10; 6 unknown), corresponding to only half of the small intestinal length of group 1 patients (*p* = 0.07). The median residual fraction of colon was 57% (min–max: 29–100%; *n* = 16) and there was no patient with a modified proximal gastrointestinal tract.

In group 3, 2 patients had a resection after ileus, 4 had extensive mucosal disease, and 1 had a motility disorder. The pathophysiological conditions of IF were 2 each with short bowel or extensive small bowel mucosal disease; and 3 patients multiple mechanisms. The resected patients had a mean small bowel length of 185 cm, and 3 patients had a modified proximal gastrointestinal tract.

### 3.2. Changes of BMI

Total of 18 patients were malnourished at FIRST with a BMI of 16.8 kg/m^2^ (13.4–18.3 kg/m^2^). They had lost weight because of their intestinal insufficiency. At LAST only 6 patients (1 group 1, 4 group 2 and 1 group 3) were malnourished with a BMI of less than 18.5 kg/m^2^ (16.2–18.3 kg/m^2^). Figure 1 shows the course of the BMI in groups 1–3 at the three time points FIRST, MAX, and LAST. The BMI at MAX was significantly higher than at FIRST in all groups. In group 1 BMI decreased significantly between MAX and LAST as a result of a reduction of PS (see below). Despite that, BMI was significantly higher at LAST than at FIRST in group 1 and group 2. Importantly, increases in BMI were not the result of overhydration or edema as these were carefully checked for clinically.

In addition, phase angle from BIA measurements and values of serum albumin were analyzed when available. Values were compared at the earliest, the maximum, and the latest time point (similar to FIRST, MAX and LAST). Phase angle was available from 22 of 50 patients (Appendix A
Figure A1). It significantly increased between the earliest time point and maximum and decreased slightly but significantly thereafter (Friedman’s test and Wilcoxon test). Serum albumin was available from 46 of 50 patients (Appendix A
Figure A1). It significantly increased between the earliest and maximal time point and decreased slightly but significantly thereafter.

### 3.3. Time Course

37 patients (22 of group 1, 11 of group 2 and 4 of group 3) were referred to us from other institutions and had received some form of PS before. At this time point (FIRST) their median time on PS was 288 days (3–6202; *n* = 37). In many of these patients PS initially had to be intensified based on insufficient urinary output or based on the course of their body weight. In these 37 patients we increased the energy support in 11, volume support in 15, and sodium support in 18 up to the values summarized at time point MAX in Table 2. At FIRST 18 of the 37 patients received an individually compounded PS; after adjustments this number increased to 22. After MAX it was possible to reduce energy support in 30/37 patients, volume support in 32/37, and sodium support in 29/37.

Total of 13 patients were newly started on PS by us (5 group 1, 5 group 2, and 3 group 3). Over the course of time we increased the energy support in 2, volume support in 3, and sodium support in 6 patients up to MAX. In 9/13 patients it was later possible to reduce the energy support and in 11/13 patients to reduce volume and sodium support.

Patients were followed for 596 days (41–2016 days) with no significant differences between the groups (Table 3).

As expected, the maximal BMI was recorded after the prescription of maximal PS. As a result of this time shift the time interval between MAX_PS_ and LAST is longer than the time interval between MAX_BMI_ and LAST (Table 3). In 10/50 patients the BMI was still increasing at LAST. In the other 40/50 patients the BMI was recorded for a median of 361 days (7–1606 days) after its maximal value, i.e., MAX_BMI_. Together the data indicate that the BMI was recorded sufficiently long after MAX_BMI_ to be regarded as stable or increasing but not further decreasing (258 days in the entire group).

### 3.4. Adjustments of Energy Support

As can be seen in Table 2 and Figure 2 energy support was already high at FIRST (24 kcal/kg/day (0–39); *n* = 50). In 13/50 patients (5 group 1, 6 group 2 and 2 group 3) energy support was increased by 9 kcal/kg/day (3–20 kcal/kg/day) to a maximum of 26 kcal/kg/day (0–39 kcal/kg/day; *n* = 50).

Because of the rebalancing of lost body weight, as a result of adaptation as well as due to the reestablishment of intestinal continuity in 11/50 patients (depicted as red symbols in Figure 2 and Appendix A
Figure A2) energy content of PS was reduced after MAX to 8.5 kcal/kg/day (0–33 kcal/kg/day; *n* = 50) at LAST (Table 2). In those patients who did not undergo reestablishment of intestinal continuity and did not get teduglutide energy support could be reduced to 13 kcal/kg/day (0–33 kcal/kg/day; *n* = 37) and it also became more uniform.

Energy support was then analyzed in group 1 and group 2 separately because of the very different small intestinal length. Group 3 was not analyzed individually because of its heterogeneous pathophysiology.

At FIRST group 1 patients had a high energy support of 26 kcal/kg/day (0–39 kcal/kg/day; *n* = 27) (Figure 2, left panel). In 5 patients energy support was increased by 13 kcal/kg/day (3–20 kcal/kg/day) resulting in a significantly higher value at MAX (28 kcal/kg/day (0–39); *n* = 27). After MAX energy support could significantly be reduced to 7 kcal/kg/day (0–30 kcal/kg/day; *n* = 27). This reduction remained significant even when we excluded 11 patients who underwent reconstructive surgery and 2 patients who received teduglutide between MAX_PS_ and LAST (9.5 kcal/kg/day (0–30); *n* = 14). Four group 1 patients without reestablishment of intestinal continuity did not need energy support at LAST; but 3 of them continued to require volume and sodium support. Only 1 patient with an ileostomy and 120 cm residual small bowel after tumor resection had achieved oral autonomy at LAST.

At FIRST group 2 patients had an energy support of 18 kcal/kg/day (0–39 kcal/kg/day; *n* = 16) (Figure 2, right panel). In 6 patients energy support was increased by 8 kcal/kg/day (4–13 kcal/kg/day) resulting in a significantly higher value at MAX (22 kcal/kg/day (0–39); *n* = 16). After MAX energy support could significantly be reduced to 14 kcal/kg/day (0–33 kcal/kg/day; *n* = 16).

Energy support consisted of amino acids, glucose, and lipids (Appendix A
Figure A2). In 13/50 patients (5 of group 1, 6 of group 2, and 2 of group 3) energy support was increased by addition of amino acids (+54% (0.41 g/kg/day;0.14–1.24); *n* = 13), by addition of glucose (+35% (0.84 g/kg/day; 0–1.97); *n* = 13) and by addition of lipids (+59% (0.44 g/kg/day;0–0.89); *n* = 13). After MAX macronutrients could be reduced in 39/50 patients (23 group 1, 11 group 2 and 5 group 3): amino acids −62% ((−0.78 g/kg/day; −0.02–−1.56); *n* = 39), glucose −73% ((−2.25 g/kg/day; −0.5–−4.03); *n* = 39), and lipids −56% ((−0.55 g/kg/day; −0.01–−1.56); *n* = 39).

### 3.5. Adjustments of Sodium and Volume Support

Dynamic adjustments of the volume and sodium content of PS are shown in Table 2 and Figure 3 and Figure 4. At FIRST volume and sodium content of PS of the entire cohort were high (volume: 1985 mL/day (527.5–7500); sodium: 128.5 mmol/day (11–728)). Especially in group 1 patients these values varied widely (Figure 3 and Figure 4). In 18/50 patients we increased PS volume by 567 mL/day (23–1625 mL/day; +22%), and in 24/50 patients we increased sodium by +91.5 mmol/day (9–237 mmol/day; +75%). Similar to energy support, volume and sodium support could significantly be reduced after MAX: volume −40% (−986 mL/day; (−0.01–−7007); *n* = 43) and sodium −63% (−140 mmol/day; (−7–−538); *n* = 40).

Because of the different pathophysiology of intestinal water and sodium losses group 1 and group 2 were then analyzed separately. Group 3 was not analyzed individually because of its heterogeneous pathophysiology.

At FIRST group 1 patients had a volume support of 2500 mL/day (625–7500 mL/day; *n* = 27) (Figure 3, left panel). In 9 patients volume support was increased by 800 mL/day (23–1625 mL/day) resulting in a significantly higher value at MAX (3000 mL/day (1500–7500); *n* = 27). After MAX volume support could significantly be reduced to 928.6 mL/day (0–3100 mL/day; *n* = 27). This reduction remained significant even when we excluded 11 patients who underwent reconstructive surgery and 2 patients who received teduglutide between MAX_PS_ and LAST (2076.5 mL/day (0–3100); *n* = 14).

At FIRST group 2 patients had a volume support of 1354 mL/day (527.5–3477 mL/day; *n* = 16) (Figure 3, right panel). In 7 patients volume support was increased by 550 mL/day (30–1500 mL/day) resulting in a significantly higher value at MAX (1500 mL/day (527.5–3477); *n* = 16). After MAX volume support could significantly be reduced to 1073.1 mL/day (0–3300 mL/day; *n* = 16).

At FIRST group 1 patients had a sodium support of 200 mmol/day (34–728 mmol/day; *n* = 27) (Figure 4, left panel). In 15 patients sodium support was increased by 103 mmol/day (9–237 mmol/day) resulting in a significantly higher value at MAX (300 mmol/day (150–728); *n* = 27). After MAX sodium support could significantly be reduced to 107 mmol/day (0–350 mmol/day; *n* = 27). This reduction remained significant even when we excluded 11 patients who underwent reconstructive surgery and 2 patients who received teduglutide between MAX_PS_ and LAST (165 mmol/day (0–350); *n* = 14).

At FIRST group 2 patients had a sodium support of 80 mmol/day (11–334 mmol/day; *n* = 16) (Figure 4, right panel). In 7 patients sodium support was increased by 137 mmol/day (14–187 mmol/day) resulting in a significantly higher value at MAX (128.5 mmol/day (21–334); *n* = 16). After MAX sodium support could significantly be reduced to 64.5 mmol/day (0–267 mmol/day; *n* = 16), significantly.

In group 1, 9 patients did not need volume or sodium support at LAST. Of these patients 8 had undergone reconstructive surgery between MAX_PS_ and LAST (red symbols).

In order to address potential total body sodium depletion and dehydration, urinary sodium concentrations were analyzed, which were available from 28 of 50 patients. Urine sodium concentration significantly increased between the earliest time point and the latest measurement (similar to FIRST and LAST; Appendix A
Figure A3) (Wilcoxon test). Of note, urine sodium concentration was initially below 20 mmol/L in 9 patients; in 7 of these urine sodium concentration increased to values above 20 mmol/L as a result of changes in the composition of PS support (Appendix A
Figure A3).

### 3.6. Relative Impact of PS Adjustments on BMI

Next we wanted to find out which of the different modifications of PS had the strongest impact on BMI. Therefore, we categorized patients with regard to increase or no change in energy, volume, and sodium support and compared the resulting groups in a pairwise fashion. In line with the analyzes depicted in Figure 2, Figure 3 and Figure 4 we included only patients of groups 1 and 2 (*n* = 43). Results are shown in Figure 5; they principally remained the same even when we added data from group 3 (not shown). Intensification of sodium support was highly significantly associated with the largest increase in BMI. Of note, the BMI at FIRST showed no significant differences between the individual groups, thus the potential for change was identical in all groups at baseline.

In line with the analysis shown in Figure 1 BMI increased under all conditions even if PS was not changed at all. This most likely reflects convalescence and adaptation as well as positive energy and nitrogen balance as a result of PS. As can be seen from the analysis of sodium and energy (Figure 5, left panel) as well as sodium and volume (Figure 5, middle panel) increases of sodium in the PS had a significantly higher impact on BMI than increases of energy or volume.

In all groups the BMI increase was significant, but the BMI increase was significantly stronger when only sodium support was intensified (Figure 5, left and middle panel, blue bars). Of note, the strong effect of increased sodium support on BMI largely occurred in patients without colon in continuity as 12 of 14 patients (86%) and 7 of 8 patients (87%) represented in the blue bars had no colon in continuity while this was the case in only 3 of 8 patients (38%) and 8 of 14 patients (57%) represented in the yellow bars.

Sodium and volume were increased concomitantly in 14 cases (Figure 5 middle panel, yellow bar). This was associated with an increase in BMI that was numerically in between *no change of sodium and volume* on the one side and *increase of sodium but not volume* on the other side (Figure 5 middle panel, green and blue bar respectively). These numerical differences were not statistically significant.

The analysis of volume and energy (Figure 5, right panel) did not yield significant differences. Of note, sole increases in the energy content of PS were rare and were associated with little changes of the BMI.

### 3.7. Reduction of Infusion Days and Weaning of PS

Over time the number of days/nights per week with PS could significantly be reduced in the entire cohort (*n* = 50) from 7 to 5 (range 0–7 days/nights per week, *p* < 0.05; Figure 6), including 13 patients, who were weaned off their PS (9 of group 1, 3 of group 2, and 1 of group 3). This was mainly achieved by reconstructive surgery, which converted 11 group 1 patients in 4 cases to type II and in 7 cases to type III anatomy. In the 39 patients without reconstructive surgery the number of infusion days per week was significantly increased between FIRST and MAX reflecting the intensification of PS. After MAX infusion days per week could significantly be reduced and 5 patients were completely weaned. Reduction of infusion days per week without reconstructive surgery was possible in 7 of 16 group 1, in 5 of 16 group 2, and in 3 of 7 group 3 patients. In 2 group 1 patients this included the application of teduglutide, which was added after MAX_PS_ (0.05 mg/kg s.c.).

## 4. Discussion

In the current study we describe the dynamic course of parenteral support and analyze its association with changes in BMI in adult patients with intestinal failure. Our cohort resembles a larger single center cohort that was assessed at a single time point, and the very large ESPEN cohort that was described in a cross sectional study and has been followed for several outcome parameters for 1 year [14,15,16]. These cohorts including our own largely represent type-I anatomy of short bowel with a substantial number of patients having multiple pathophysiological mechanism for IF. Length of the remaining small intestine and of the remaining colon were very similar to the cohort from Copenhagen [14].

Patients referred to us received PS for a shorter period of time (288 days; (3–6202)) than the Copenhagen cohort at their cross-sectional evaluation (mean duration of PS 6.3 years, 11% <6 month; 39% <2 years) or the ESPEN cohort (26% on PS for <1 year) [15,16]. Thus, together with the patients who were started on PS by us, our cohort represents early IF at the time point FIRST and our analysis addresses the early dynamics of parenteral support. In line with that volume and sodium support were similar in our cohort at LAST to that reported from the Copenhagen cohort [14]. Sodium support was not reported in the ESPEN surveys [15,16].

We observed significant and clinically meaningful changes in BMI between time points FIRST and MAX as well as FIRST and LAST. To our knowledge this is the first detailed description of the dynamics of PS and the related course of BMI in early adult IF. In this phase the BMI reflects PS as a modifiable factor as well as convalescence and adaptation which may both be secondarily affected by PS. Because of these three factors BMI increased between FIRST and LAST in groups 1 and 2. In group 3 BMI increased only between FIRST and MAX, probably because the group was small, more heterogeneous, and had less potential for prolonged adaptation.

We modified the energy, volume, and sodium content of PS based on recommended clinical parameters between FIRST, MAX, and LAST. These modifications were associated in a timely manner with changes in the BMI. The time courses of phase angle and serum albumin, although not available from all patients, indicate that the increased BMI was not a result of overhydration. Because of stomal and diarrheal losses volume support is regarded especially important in the treatment of IF [3,15,16]. On the other hand, energy content varies because of the individual ability to absorb macronutrients, because of individual energy requirements, and because of the individual BMI [15]. Study protocols for pharmacologic treatment of IF also address reduction of PS volume as the primary study end point [7]. Because of the fixed sodium concentration of small intestinal chyme of about 100 mmol/L, voluminous intestinal losses translate to high sodium losses especially in patients with a small bowel stoma, i.e., with no colon in continuity [17].

In line with the recommendation to use urine output to guide volume support PS volume was already high in the majority of patients at FIRST. While sodium support was also high in many but not all patients at FIRST, intensifications of the sodium support were the most frequent and the relatively most extensive modifications of the PS between FIRST and MAX_PS_. Also, sole increases in sodium support predominantly occurred in patients without colon in continuity. In order to address the impact of energy, volume, and sodium support on BMI between FIRST and MAX_BMI_ we used the Kruskal-Wallis test followed by Mann-Whitney U-test. Intensification of sodium support was associated with a significantly larger increase of BMI than intensification of energy support indicating that sodium had an independent effect on the recovery of BMI. Ultimately, these findings indicate that patients with sufficient energy support may be anabolically stimulated by additional sodium support.

Our findings are in line with two small studies in children with large intestinal stoma losses, which have shown that total body sodium depletion precludes physiological weight gain, and that correction of sodium depletion facilitates weight gain [18,19,20]. In adults the recommendations regarding the quantity of sodium support have theoretically been derived from estimated sodium losses. Our data add evidence, that vigorous sodium support in the early phase of IF is accompanied by an independent, clinically meaningful beneficial effect on the recovery of BMI. As the physiological mechanism increased sodium support facilitates normalization of volume status in dehydrated patients disinhibiting low volume status as a potential katabolic factor or may act as the basis for anabolic metabolism in euvolemic patients through yet unknown mechanisms [21]. In this context it is noteworthy that sodium restriction impairs insulin secretion, which is an anabolic factor [22], and that cell swelling as a result of normalized volume status acts as an anabolic factor [23].

Total body sodium is poorly reflected by serum sodium and may only be detected by measuring urine sodium excretion or alterations of the renin-angiotensin-aldosterone axis [24]. While we used clinical parameters (and urinary sodium in some patients) to modify sodium support, future studies will have to address how sodium support is precisely optimized similar to the approach taken with regard to volume in the phase II and III trials testing GLP-2 analogues [7,25,26].

We are aware that in the referred patients postoperative convalescence may have already occurred to some degree. In addition, there were patients in the three groups who were affected by their underlying disease for prolonged periods of time before the initiation of PS. Their intestines may already have adapted to the small bowel situation to some degree. Also, 7 out of 16 patients with Crohn’s disease received specific medical therapy, which may have had a positive effect of its own. Despite that, intensified sodium support was strongly associated with an increase in BMI, supporting the notion that sodium acts as an independent factor even under these conditions.

After MAX_PS_ PS could be reduced both with regard to sodium and volume as well as energy. This was associated with a significantly lower BMI at LAST in group 1 patients but with no significant decrease but a preserved increase in BMI in group 2 patients. Eventually energy support was somewhat less in our patients at LAST than in the Copenhagen cohort [14] and in the ESPEN survey [15]. The most likely explanation for this is that deficits were replenished, and a new equilibrium was achieved. It may be speculated that the positive effect of sodium on BMI also translated on intestinal function by virtue of enterocyte number and/or function and thus stimulated intestinal adaptation. Importantly patients who changed from type I to type II anatomy acquired their colon not only for the absorption of electrolytes and water but also for energy absorption and improved endocrine regulation of absorption and growth of small intestinal villi [27].

Type I short bowel anatomy has been shown to have comparably little potential for adaptation to oral autonomy [2,11]. Nevertheless, reduction of the intensity of PS is of clinical importance with regard to quality of life and frequency of complications [16,28]. In our cohort patients without reconstructive surgery had a similar weaning rate as patients in the ESPEN cohort [16]. In addition, our data show that even in those group 1 patients who did not undergo reconstructive surgery and who did not achieve oral autonomy the number of days with PS per week was significantly reduced.

## 5. Conclusions

In summary we show that significant dynamic adjustments of PS are required during the early course of adult IF. Vigorous sodium support, which may initially be underestimated, has the strongest impact on the recovery of BMI and acts as an independent factor. Finally reconstructive surgery and intestinal adaptation allow reduction and, in some cases, weaning of PS over time.

## Figures and Tables

**Figure 1 nutrients-12-03426-f001:**
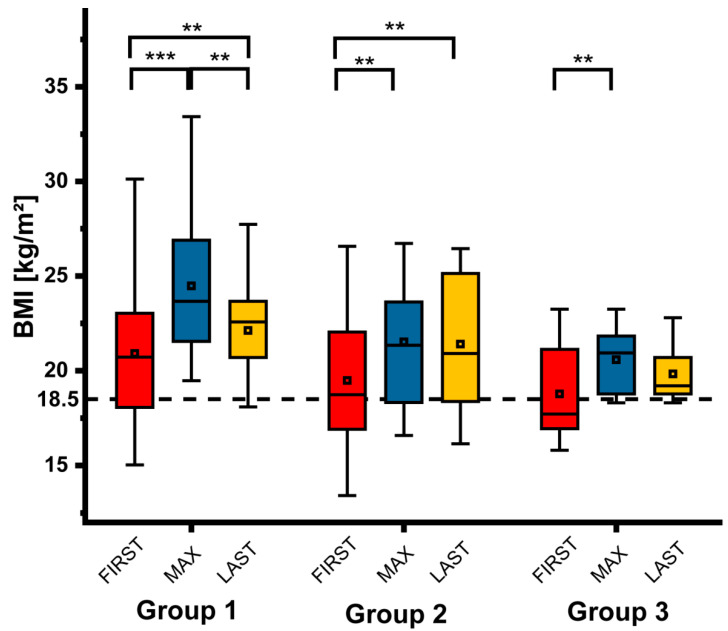
Time course of the BMI in the different groups (group 1: small bowel enterostomy, group 2: jejuno-colic anastomosis, group 3: jejuno-ileocolic anastomosis or no bowel resection). At LAST only patients without reconstructive surgery and without teduglutide are included (** *p* < 0.01; *** *p* < 0.001).

**Figure 2 nutrients-12-03426-f002:**
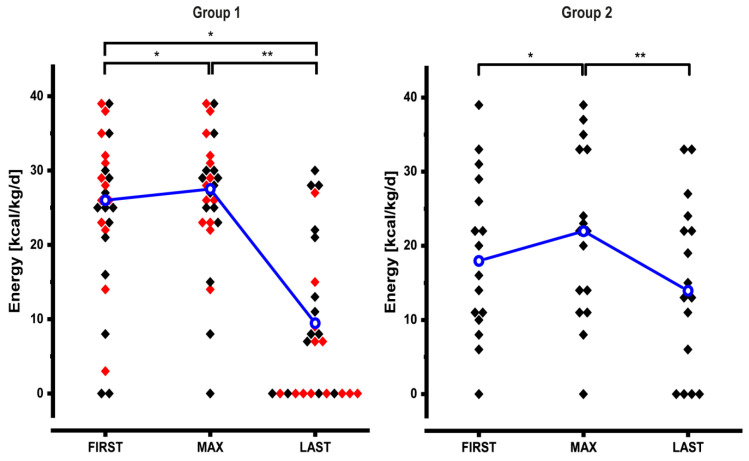
Dynamic changes of energy support in group 1 and group 2 patients. In group 1, 11 patients had reconstructive surgery and 2 patients received teduglutide between MAX and LAST, they are indicated as red symbols and they are not included in the statistical analysis (median values and significant differences) at time point LAST. Median values are connected by the blue line. See text for details (* *p* < 0.05; ** *p* < 0.01).

**Figure 3 nutrients-12-03426-f003:**
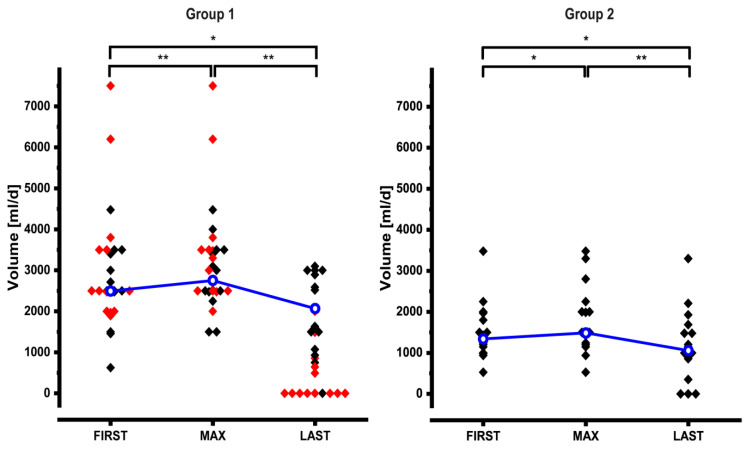
Dynamic changes of volume support in group 1 and group 2 patients. In group 1, 11 patients had reconstructive surgery and 2 patients received teduglutide between MAX and LAST, they are indicated as red symbols and they are not included in the statistical analysis (median values and significant differences) at time point LAST. Median values are connected by the blue line. See text for details (* *p* < 0.05; ** *p* < 0.01).

**Figure 4 nutrients-12-03426-f004:**
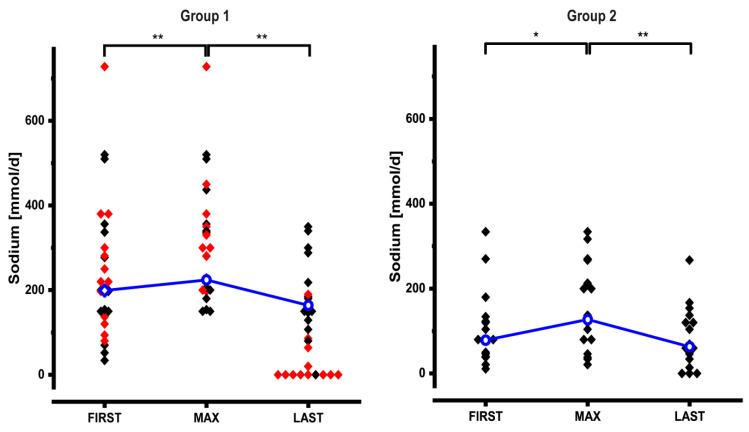
Dynamic changes of sodium support in group 1 and group 2 patients. In group 1, 11 patients had reconstructive surgery and 2 patients received teduglutide between MAX and LAST, they are indicated as red symbols and they are not included in the statistical analysis (median values and significant differences) at time point LAST. Median values are connected by the blue line. See text for details (* *p* < 0.05; ** *p* < 0.01).

**Figure 5 nutrients-12-03426-f005:**
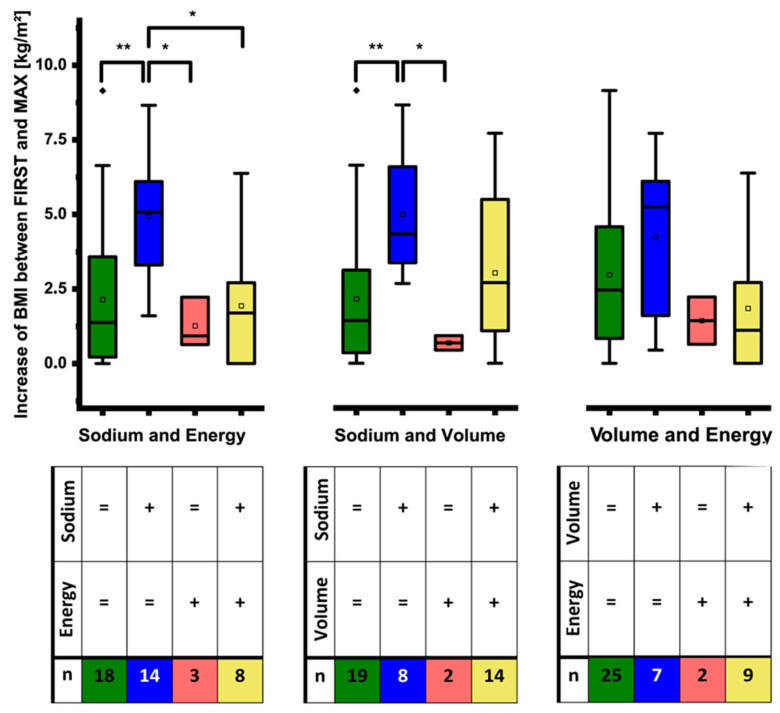
Impact of modifications of energy, volume, and sodium support on the increase in BMI between FIRST and MAX_BMI_. The table indicates if one of the two considered components of the parenteral support was intensified (+) or not (=) and how many patients were affected (groups 1 and 2; *n* = 43 in each panel). Intensification of only sodium support was highly significantly associated with the largest increase in BMI (blue boxes left and middle panel). These changes in sodium were the most frequent and the relatively most extensive. They mainly occurred in patients without colon in continuity. Of note, at FIRST there were no significant differences in the absolute BMI between the groups (not depicted in the figure) (* *p* < 0.05; ** *p* < 0.01).

**Figure 6 nutrients-12-03426-f006:**
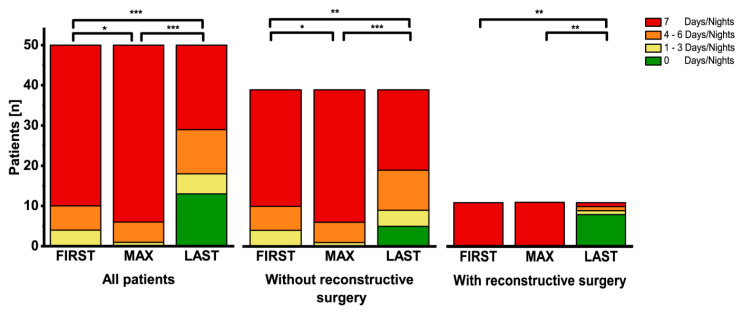
Number of infusion days/nights per week. Intensification of PS resulted in a significantly higher number of infusion days/nights per week at MAX compared to FIRST. Reconstructive surgery facilitated weaning or significant reduction of PS days/nights in the majority of patients (right panel). In 39 patients without reconstructive surgery the number of infusion days/nights per week was significantly increased between FIRST and MAX. After MAX infusion days/nights per week could significantly be reduced and 5 patients were completely weaned. Reduction of infusion days/nights per week without reconstructive surgery was possible in 7 of 16 group 1, in 5 of 16 group 2, and in 3 of 7 group 3 patients. In 2 group 1 patients this included the application of teduglutide, which was added after MAX_PS_ (0.05 mg/kg s.c.). (* *p* < 0.05; ** *p* < 0.01; *** *p* < 0.001).

**Table 1 nutrients-12-03426-t001:** Demographics of the study population and of the groups defined by their functional anatomy. Median (min—max).

	All	Group 1 Small Bowel Enterostomy	Group 2 Jejuno-Colic Anastomosis	Group 3 Jejuno-Ileocolic Anastomosis/No Surgery
N	50	27	16	7
male/female	24/26	16/11	5/11	3/4
Age (years)	53 (21–80)	53 (21–73)	51.5 (34–80)	62 (31–80)
Pathophysiology				
short bowel (SB)	27 (54%)	15	10	2
mechanical obstruction (MO)	1 (2%)	0	1	0
extensive small bowel mucosal disease (ESBMD)	2 (4%)	0	0	2
multiple mechanisms:	20 (40%)	12	5	3
SB + Intestinal fistula	9	7	2	0
SB + Intestinal fistula + MO	1	1	0	0
SB + MO	5	2	1	2
SB + ESBMD	4	2	2	0
SB + MO + ESBMD	1	0	0	1
Small intestine (cm)	100 (0–240), *n* = 28	100 (0–240), *n* = 17	55 (10–170), *n* = 10	185 (185), *n* = 1
Colon (%)	71 (0–100), *n* = 43	potentially 72 (29–100) *n* = 17; 3 no remaining colon; 7 unknown	57 (29–100), *n* = 16	100 (70–100), *n* = 7
Modified proximal gastrointestinal tract (*n*)	6	3	0	3
Reconstructive surgery (*n*)	11	11 (4 to type II; 7 to type III)	0	0

**Table 2 nutrients-12-03426-t002:** Composition of parenteral support (PS) at time points FIRST, MAX, and LAST in the entire cohort. For the time point FIRST the cohort is stratified into patients who were newly started on PS and patients who were referred with existing PS. Median (min–max).

		FIRST		MAX	LAST
	All (*n* = 50)	Newly Started PS (*n* = 13)	Existing PS (*n* = 37)		
**BMI** **(kg/m^2^)**	20.3 (13.4–30.1)	19.3 (13.4–23.2)	20.4 (18.9–30.1)	22.4 (16.6–33.4)	21.9 (16.2–36)
**Patients with BMI < 18.5 kg/m^2^** **(n)**	18	6	12	5	6
**energy** **(kcal/kg/day)**	24 (0–39)	22 (0–35)	25 (0–39)	26 (0–39)	8.5 (0–33)
**amino acids** **(g/kg/day)**	0.97 (0–1.87)	0.94 (0–1.56)	1.1 (0–1.87)	1.12 (0–1.87)	0.39 (0–1.53)
**glucose** **(g/kg/day)**	2.83 (0–5)	2.35 (0–3.85)	2.98 (0–5)	3.03 (0–5.26)	0.96 (0–4.46)
**lipids** **(g/kg/day)**	0.89 (0–1.56)	0.86 (0–1.56)	0.9 (0–1.49)	0.95 (0–1.56)	0.37 (0–1.45)
**volume** **(mL/day)**	1985 (528–7500)	1500 (845–4477)	2000 (528–7500)	2477 (528–7500)	1000 (0–3300)
**sodium** **(mmol/day)**	128.5 (11–728)	134 (34–510)	120 (11–728)	203 (21–728)	74.5 (0–350)
**days/week**	7 (2–7)	7 (3–7)	7 (2–7)	7 (3–7)	5 (0–7)

**Table 3 nutrients-12-03426-t003:** Time course as assessed by FIRST, MAX, and LAST. Median (min–max).

	All	Group 1 Small Bowel Enterostomy	Group 2 Jejuno-Colic Anastomosis	Group 3 Jejuno-Iieo-Colic Anastomosis/No Surgery
**N**	50	27	16	7
Follow up (day)	596 (41–2016)	637 (91–1995)	679 (41–2016)	505 (77–1310)
Time of disease before first PS (day)	92 (0–11733)	92 (0–8036)	92 (0–8544)	91 (0–11733)
PS before FIRST (day) ^1^	288 (3–6202) *n* = 37	287.5 (10–5108) *n* = 22	409 (0–6202) *n* = 11	490 (26–1162) *n* = 4
FIRST until MAX (day) ^2^				
PS ^3^	18 (0–982)	28 (0–982)	18 (0– 447)	0 (0–302)
BMI ^4^	236 (0–1806)	280 (0–1645)	172 (0–1806)	147 (0–365)
MAX until LAST (day) ^2^				
PS ^3^	512.5 (0–1864)	507 (0–1864)	631.5 (0–1829)	463 (77–1008)
BMI ^4^	253.5 (0–1606)	203 (0–1606)	283.5 (0–826)	140 (0–1310)

^1^ 37 patients were referred from other centers and already received PS for various time periods (referred as PS before FIRST). All other data include these 37 patients plus the 13 patients who were newly put on PS. ^2^ In some patients values of maximal PS or maximal BMI were already recorded at time point FIRST resulting in a time interval of 0 days between FIRST and MAX in these cases. The same was true in some cases between MAX and LAST. ^3^ If energy, volume, or sodium were modified at different times the average time point was used. ^4^ Maximal BMI was recorded after the prescription of maximal PS; as a result of this time shift the time interval between maximal PS and LAST is longer than the time interval between maximal BMI and LAST.

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
