# Peer review of "Dynamic Adjustments of Parenteral Support in Early Adult Intestinal Failure—Essential Role of Sodium"

_nutrients, 2020, doi:10.3390/nu12113426_

Round 1

Reviewer 1 Report

Typographical and Grammatical errors: 

  • Page 2, Line 89: ‘the prescriptions issued for the compounding pharmacy’., should perhaps be: issued by the compounding… 
  • Page 10, line 273: ‘In group1 9 patients’…; should be: In group 1, 9 patients 

Comments: 

  • In the methodology and description of the general support of the patient besides the PS, did any of these patients have quantitated (objective measurement) analysis of the intestinal and / or urinary loses of Na or other electrolytes?  
  • Did any of these patients also received hydration solutions to replace loses or were those included in the PS volume and composition? 
  • The comment on lines 150 – 152 of the results that the BMI increases were not due to over hydration or fluid retention, are very important and should also be in mentioned or included in the ‘Patients’ section, perhaps even with emphasis in those patients who may have had low serum Albumin levels, particularly those with possible Protein-losing enteropathy from IBD (Crohn’s)  
  • The use of Teduglutide in some patients should also be mentioned in the ‘Patients’ section, together with other medications patients received. 
  • Perhaps a few words or mention of whether patients with IBD (Crohn’s) were disease-free after surgical resection or if not, as it may be frequently the case, whether they also had or were receiving therapy for that condition, which may also have had an impact in the improvement shown.  

Author Response

We thank the reviewer for the helpful comments.

Point by point response:

Typographical and Grammatical errors have been corrected. Line 95 and 289

Did any of these patients also received hydration solutions to replace loses or were those included in the PS volume and composition? 

“The reported values summarize the entire PS.” This has been added to line 96.

All reviewers raise the question how overhydration was ruled out as the cause of the improved BMI.

We have gathered and analysed data from BIA measurements, serum albumin levels and urinary sodium concentrations when available. These data are described in paragraph “Changes of BMI” (line 166 – 172) and “Adjustments of sodium and volume support “ (line 291 – 296) and are summarized in 2 supplementary figures. We would like to put this into suppl. figures because we do not have data from the entire cohort. These additional data 1) rule out overhydration, 2) prove that BMI increase was accompanied by an improved phase angle and 3) show that some patients were dehydrated as indicated by their low urinary sodium. This is taken up in the discussion to support our interpretation that sodium is an anabolic factor. At the same time, we have phrased more carefully such that normalization of volume status by increased sodium support may act as the basis for anabolic metabolism or by disinhibiting low volume status as a katabolic factor.

… perhaps even with emphasis in those patients who may have had low serum albumin levels, particularly those with possible Protein-losing enteropathy from IBD (Crohn’s) …

… Perhaps a few words or mention of whether patients with IBD (Crohn’s) were disease-free after surgical resection or if not, as it may be frequently the case, whether they also had or were receiving therapy for that condition, which may also have had an impact in the improvement shown.  

7 out of 16 patients with Crohn’s disease received CD specific medical therapy, which was stable over time except for one patient. It is difficult and, in some cases, impossible to differentiate between active CD and intestinal failure. We have taken that problem up as a weakness of our study. Line 423 – 425.

The use of Teduglutide in some patients should also be mentioned in the ‘Patients’ section, together with other medications patients received. 

Teduglutide was added to the medical treatment in 2 patients after MAX, i.e. after their PS had been optimized. This has been added both in the “Patients” section (line 89) as well as in the “Results” section (line 346). Thus, their Teduglutide treatment does not influence the analysis between FIRST and MAX. Therefore, we have not analysed them separately.

Reviewer 2 Report

This paper relates the course for BMI development to the optimization of fluid/electrolyte and nutritional treatment in 50 patients. The mean duration to the first visit is 288 days and the cohort consists of a mix of patients referred to Rostock for optimizing of treatment and novel own patients. The course may be influenced both by the longterm adaptation after the extensive intestinal resections and the optimization of treatment, which the authors are of course aware of and consider in the discussion.

An interesting finding is that data suggest that optimal hydration/sodium supply has in itself an anabolic effect. The authors might expand the discussion on line 372-376 to include possible mechanisms behind such an effect, Eg there are reports. mainly in diabetological litterature, that dehydration and hyponatremia impairs insulin secretion and or insulin sensitivity (eg Luther et al J Endocrinol Met 2014; 99:2014-2022) and thereby may have an antianabolic effect. Some patients receive increased sodium support without increase in volume support. Was that decision based on plasma Na measurement or was Na and K and K/Na ratio in urine measured? 

 Line 15-152: "Importantly increases in BMI were not the result of overhydration or edema as these were carefully checked for clinically". Could there be a variation in degree of hydration without obvious signs of overhydration?  

Concommitttant medication: A couple of patients have very high stomal losses, replacement with up to 7,5 liter and 728 mmol Na/d. Did these patients take their proton pump inhibitor intravenously?

Details: In the box below fig 5 the = sign should be replaced with a - sign to avoid confusion. Line 71: "trails" should be "trials". The introduction of "FIRST, MAX, LAST" in the abstract makes it more difficult for the abstract reader, let be not for the reader who reads the whole paper.

Author Response

We thank the reviewer for the helpful comments.

Point by point response:

An interesting finding is that data suggest that optimal hydration/sodium supply has in itself an anabolic effect. The authors might expand the discussion on line 372-376 to include possible mechanisms behind such an effect, Eg there are reports. mainly in diabetological litterature, that dehydration and hyponatremia impairs insulin secretion and or insulin sensitivity (eg Luther et al J Endocrinol Met 2014; 99:2014-2022) and thereby may have an antianabolic effect.

We thank the reviewer for this helpful suggestion. We have added a sentence referring to the catabolic effect of sodium restriction mediated by decreased insulin secretion as well as the effect of cell swelling as an anabolic signal (including key references). Line 411 – 413.

Some patients receive increased sodium support without increase in volume support. Was that decision based on plasma Na measurement or was Na and K and K/Na ratio in urine measured? 
AND
 Line 15-152: "Importantly increases in BMI were not the result of overhydration or edema as these were carefully checked for clinically". Could there be a variation in degree of hydration without obvious signs of overhydration?  

We have gathered and analysed data from BIA measurements, serum albumin levels and urinary sodium concentrations when available. These data are described in paragraph “Changes of BMI” (line 166 – 172) and “Adjustments of sodium and volume support “ (line 291 – 296) and are summarized in 2 supplementary figures. We would like to put this into suppl. figures because we do not have data from the entire cohort. These additional data 1) rule out overhydration, 2) prove that BMI increase was accompanied by an improved phase angle and 3) show that some patients were dehydrated as indicated by their low urinary sodium. This is taken up in the discussion to support our interpretation that sodium is an anabolic factor. At the same time, we have phrased more carefully such that normalization of volume status by increased sodium support may act as the basis for anabolic metabolism or by disinhibiting low volume status as a katabolic factor.

Concommitttant medication: A couple of patients have very high stomal losses, replacement with up to 7,5 liter and 728 mmol Na/d. Did these patients take their proton pump inhibitor intravenously?

Detail and a reference have been added to the sentence: All patients received proton pump inhibitors and in cases of very high stomal output this was given intravenously. Line 86.

Details: In the box below fig 5 the = sign should be replaced with a - sign to avoid confusion.

We would like to keep the “=” sign because the “-“ sign may be confused with a reduction of the respective compound of PS.

Line 71: "trails" should be "trials".

The typo has been corrected. Line 72.

The introduction of "FIRST, MAX, LAST" in the abstract makes it more difficult for the abstract reader, let be not for the reader who reads the whole paper

The technical terms FIRST, MAX and LAST are explained in the abstract and in the main document. Specifically, the abstract is limited to 200 words, therefor we had to go with FIRST, MAX and LAST.

Reviewer 3 Report

Dear authors thank for the effort in describing dynamic changes in parenteral support. 

However I have several major concerns. 

GENERAL CONCERN: I am a bit skeptical in an article with very strong conclusions over sodium intake and without any data of urinary sodium output. 

ABSTRACT 

Phrase starting at line 27 in very unclear and need to be rephrase by native English speaker. 

Conclusion that sodium acts as an independent anabolic factor is not supported neither by results of this study nor by strong literature references. 

INTRODUCTION

The whole introduction needs a native English speaker editing. Especially the paragraph starting from line 47 is very unclear. 

PATIENTS:

Why all patients received PPI?

Why it is not mentioned that 2 patients received Teduglutide treatment? Those patients are not comparable with the remnant cohort and, in my opinion, they should be excluded from the analysis. 

DEMOGRAPHICS: 

Please specify which are the multiple pathophysiological mechanism of intestinal failure of the many of the patients (20). 

CHANGES IN BMI

Line 145 I would erase only because you are presenting results. 

Line 151: "Importantly, increase of BMI were not a results of over hydration or edema as these were carefully checked for clinically": this a crucial point. I think this could not be expressed, because, especially in type 1 short bowel changes in hydration are often not clinically relevant because they are an expression of a previous dehydration. Furthermore motility disorders can be complicated by third space oedema which is not quantifiable clinically. It is mandatory to corroborate this with some laboratory finding (Albumin, Urinary sodium, Urea, Creatinin).

TABLE 3: not informative, can be deleted. 

FIGURE 5: Increase in BMI from FIRST to MAX is this MAX BMI or MAX PS? Furthermore difficult to drive conclusion since the group of sodium increase without energy increase in very scarce only 3 patients and without volume increase even smaller 2 patients. I would be more cautious in conclusions. 

REDUCTION OF INFUSION DAYS AND WEANING: I am very surprised that 2 patients treated with Teduglutide are presented the first time at page 12. Teduglutide is a treatment which impact deeply on PS requirements. Those patients should be either excluded either object of a separate analysis in all segments. 

DISCUSSION: overall the discussion is very week of argument. Literature research was not made properly and many references are missing. In particular please read carefully and cite a recent paper published on the importance of colon continuity in PS: "The colon as an energy salvage organ for children with short bowel syndrome. Norsa L, Lambe C, Abi Abboud S, Barbot-Trystram L, Ferrari A, Talbotec C, Kapel N, Pigneur B, Goulet O.Am J Clin Nutr. 2019 Apr 1;109(4):1112-1118. doi: 10.1093/ajcn/nqy367". 

I'm also very skeptical on your conclusion on the independent anabolic affect of sodium intake. This conclusion is absolutely not supported by your results and cannot be corroborate by literature. I think that changes in BMI due to sodium are probably more attribuable to hydric balance in the body even in the absence of clinical edema. 

Line 399: this sentence is too strong and not at all supported by your results and there are no references to explain this sentence. I do feel that the impact of reconstructive surgery is far stonger than the simple adaptation and is not discussed with appropriate references. 

Author Response

We thank the reviewer for the helpful comments.

Point by point respons:

The whole introduction needs a native English speaker editing. Especially the paragraph starting from line 47 is very unclear.

The entire manuscript has been read and edited by a native English speaker. The paragraph starting at line 48 has been rephrased to make it clearer.

Why all patients received PPI?

Patients received PPI because it reduces gastric/intestinal secretion. The reference has been included in the text: Jeppesen, P. B.; Staun, M.; Tjellesen, L.; Mortensen, P. B. (1998): Effect of intravenous ranitidine and omeprazole on intestinal absorption of water, sodium, and macronutrients in patients with intestinal resection. In: Gut 43 (6), S. 763–769. DOI: 10.1136/gut.43.6.763. Line 86.

Why it is not mentioned that 2 patients received Teduglutide treatment? Those patients are not comparable with the remnant cohort and, in my opinion, they should be excluded from the analysis.
AND
I am very surprised that 2 patients treated with Teduglutide are presented the first time at page 12. Teduglutide is a treatment which impact deeply on PS requirements. Those patients should be either excluded either object of a separate analysis in all segments. 

Teduglutide was added to the medical treatment in 2 patients after MAX, i.e. after their PS had been optimized. This has been added both in the “Patients” section (line 89) as well as in the “Results” section (line 346). Thus, their Teduglutide treatment does not influence the analysis between FIRST and MAX. Therefore, we have not analysed them separately.

Phrase starting at line 27 in very unclear and need to be rephrase by native English speaker. 

A full stop was missing and has been added: à “Analysis of variance was used to test the relative impact of changes in energy, volume or sodium. 50 patients were followed for 596 days.” (Line 28)

Please specify which are the multiple pathophysiological mechanism of intestinal failure of the many of the patients (20).

The demographics table (table 1) has been extended to include this detail.

Line 145 I would erase only because you are presenting results. 

We agree that line 145 does not belong to the results section. Nevertheless, it is important to specify the principle intent of PS. It has therefore been moved from the “Results” section to the “Patients” section. Line 79.

TABLE 3: not informative, can be deleted. 

Reviewer #1 and #2 did not criticise table 3. We believe that it is important to demonstrate the dynamics along the time axis. We would like to keep table 3.

FIGURE 5: Increase in BMI from FIRST to MAX is this MAX BMI or MAX PS? Furthermore difficult to drive conclusion since the group of sodium increase without energy increase in very scarce only 3 patients and without volume increase even smaller 2 patients. I would be more cautious in conclusions. 

It is MAX-BMI, this has been added. Line 310.

There must be a misunderstanding. Group “sodium increase without energy” is 14 patients and group “sodium without volume increase” is 8 patients. The other groups are “energy or volume increased but sodium remained constant”. These groups are small because this scenario is comparably rare. Nevertheless, the conclusion is derived from the analysis of those patients who had sodium increased only and thus corroborates the special importance of sodium.

All reviewers raise the question how overhydration was ruled out as the cause of the improved BMI.

We have gathered and analysed data from BIA measurements, serum albumin levels and urinary sodium concentrations when available. These data are described in paragraph “changes of BMI” (line 166 – 172) and “Adjustments of sodium and volume support “ (line 291 – 296) and are summarized in 2 supplementary figures. We would like to put this into suppl. figures because we do not have data from the entire cohort. These additional data 1) rule out overhydration, 2) prove that BMI increase was accompanied by an improved phase angle and 3) show that some patients were dehydrated as indicated by their low urinary sodium. This is taken up in the discussion to support our interpretation that sodium is an anabolic factor. At the same time, we have phrased more carefully such that normalization of volume status by increased sodium support may act as the basis for anabolic metabolism or by disinhibiting low volume status as a katabolic factor.

Round 2

Reviewer 3 Report

Thanks to the authors for this revised version of the manuscript.

I still have two major concerns in the paper:

  1. You answer that Tedoglutide was added after MAX. This reinforce my feeling that those 2 patients must be taken out from the analysis because some of your conclusions are also based on the concept of intestinal adaptation between  MAX and LAST which in those 2 patients with type 1 short bowel was stronghly mediated by Teduglutide treatment.
  2. Please remove from the abstract the concept that sodium in an independent anabolic factor. This cannot be drawn with certitude by your results, since no functional study on the impact of sodium on any anabolyc mechanism was made.

Furthermore please state limitations of the study such as the retrospective design of the study and the eteregenous population. Some patients are referred to your center immediately at PN start Others arrives later on when the process of intestinal adaptation has allready begun. Some patients received intestinal recontruction while other did not. Timing of intestinal reconstruction are very different. Some patients are monitored with urinary  sodium output while other not.

Author Response

We thank the reviewer for the comments.

Point by point response to reviewer #3:

You answer that Teduglutide was added after MAX. This reinforce my feeling that those 2 patients must be taken out from the analysis because some of your conclusions are also based on the concept of intestinal adaptation between  MAX and LAST which in those 2 patients with type 1 short bowel was strongly mediated by Teduglutide treatment.

The two patients have been taken out of the analysis like the 11 patients who underwent reconstructive surgery. This is explained in the Methods section (line 104 – 107) and is repeated in the figure legends. (Figure 1, 2, 3, 4 and suppl. Figure 2)

Please remove from the abstract the concept that sodium in an independent anabolic factor. This cannot be drawn with certitude by your results, since no functional study on the impact of sodium on any anabolic mechanism was made.

The abstract has been modified accordingly. Line 33.

Furthermore please state limitations of the study such as the retrospective design of the study and the heterogenous population. Some patients are referred to your center immediately at PN start Others arrives later on when the process of intestinal adaptation has allready begun. Some patients received intestinal reconstruction while other did not. Timing of intestinal reconstruction are very different. Some patients are monitored with urinary sodium output while other not.

These are new comments, that were not raised in the first review. All of these details are already clearly pointed out in the manuscript in their respective context. We do not believe that an extra paragraph repeating these will add new understanding or new interpretation.